# Failure without Tears: Two-Step Attachment in a Climbing Cactus

**DOI:** 10.3390/biomimetics8020220

**Published:** 2023-05-25

**Authors:** Nick P. Rowe, Lily Cheng Clavel, Patricia Soffiatti

**Affiliations:** 1AMAP, University of Montpellier, CIRAD, CNRS, INRAE, IRD, 34398 Montpellier, France; 2Department of Botany, Federal University of Parana State (UFPR), Curitiba CEP81531-990, Brazil

**Keywords:** climbing cactus, spine and root strength, two-step attachment, benign failure, soft robotic technologies, slow-fast processes

## Abstract

Climbing plants can be extremely adaptable to diverse habitats and capable of colonising perturbed, unstructured, and even moving environments. The timing of the attachment process, whether instantaneous (e.g., a pre-formed hook) or slow (growth process), crucially depends on the environmental context and the evolutionary history of the group concerned. We observed how spines and adhesive roots develop and tested their mechanical strength in the climbing cactus *Selenicereus setaceus* (Cactaceae) in its natural habitat. Spines are formed on the edges of the triangular cross-section of the climbing stem and originate in soft axillary buds (areoles). Roots are formed in the inner hard core of the stem (wood cylinder) and grow via tunnelling through soft tissue, emerging from the outer skin. We measured maximal spine strength and root strength via simple tensile tests using a field measuring Instron device. Spine and root strengths differ, and this has a biological significance for the support of the stem. Our measurements indicate that the measured mean strength of a single spine could theoretically support an average force of 2.8 N. This corresponds to an equivalent stem length of 2.62 m (mass of 285 g). The measured mean strength of root could theoretically support an average of 13.71 N. This corresponds to a stem length of 12.91 m (mass of 1398 g). We introduce the notion of two-step attachment in climbing plants. In this cactus, the first step deploys hooks that attach to a substrate; this process is instantaneous and is highly adapted for moving environments. The second step involves more solid root attachment to the substrate involving slower growth processes. We discuss how initial fast hook attachment can steady the plant on supports allowing for the slower root attachment. This is likely to be important in wind-prone and moving environmental conditions. We also explore how two-step anchoring mechanisms are of interest for technical applications, particularly for soft-bodied artefacts, which must safely deploy hard and stiff materials originating from a soft compliant body.

## 1. Introduction

Safe and reliable attachment is a key process for living organisms and of great interest for new technologies and materials science. Attachment is especially important for climbing plants, allowing them to attach to trees and search for light without the need for developing large trunks. Different attachment mechanisms have evolved many times in climbing plants from spines and hooks to highly adaptable structures capable of movement by growth such as twining stems, tendrils, and adhesive roots.

Many of these functional aspects of climbing plants are becoming of interest for developing new technologies and new materials for soft robotics and other bio-inspired applications [1,2,3]. Aspects such as additive growth mimicking growth in length of climbing plants [4,5,6], adaptive orientated stem movements [7,8,9], hook and micro-hook attachment [10,11,12], and mobile searcher artefacts [13,14,15] are all examples of climbing-plant-inspired technologies that have co-opted specific biological traits for technical functionalities. Some of these technical innovations have not only reproduced specific tasks but have also included an adaptive or strategic functionality that can behave variably according to a changing local environment. So, today, the search for bio-inspired technologies and principles is not only seeking for specific single functions but also the adaptive principles and strategies that can control or make choices about deploying specific functions and tasks.

### 1.1. Adaptive Strategies Using Multi-Attachment Mechanisms

Climbing plants are superb examples of organisms that form safe attachments in heterogeneous environments and when perturbation such as wind and rain can easily interrupt or prevent successful anchorage [16]. Climbing plants are remarkably diverse in terms of their kinds of attachment [17] but also how they combine modes of attachment and develop “steps” during attachment to optimise safe anchorage for a diversity of substrates and environmental conditions. The ability to efficiently form attachment on variable substrates and under diverse conditions is of immense biological importance for ecological success and survival. These adaptive features also represent a holy grail for new technologies and artefacts aiming to form safe and reliable attachment under variable and unpredictable conditions [18].

For example, in Velcro-like connections, anchorage cannot occur if either of the hook or loop surfaces is absent or damaged [19]. Similarly, the contact adhesion of gecko-like adhesion depends on weak intermolecular van der Waal’s forces, which can fail if the surface becomes modified in terms of water content or contaminated [20]. Some climbing plants rely on only one kind of attachment mechanism such as the active tendrils of *Vitis vinifera* (Vitaceae). However, it is becoming increasingly clear that many other climbing plants can combine different mechanisms during their lifetime [21,22]. This has the advantage of providing a safer “choice” of mechanisms for ensuring attachment when faced with different substrates and when environmental conditions might show sudden and potentially catastrophic changes. Some tropical lianas can initiate a weak initial contact via leaf friction and then secure a far stronger attachment via angled branches [23].

For instance, an initial anchorage process might be performed by a certain form of attachment, providing a “quick” fixation, or bracing, during initial contact that keeps the plant stem in place against the support while other slower growth mediated process such as “twining around” has time to complete one or two turns around the support. There is an initial passive contact and anchorage, which is instantaneous on contact, and then a slower more permanent attachment by twining, which relies on stem growth. Recent studies have shown that many twining plants of the tropical rainforest use micro-spines on the apical twining leading shoots [24]. These can ensure high friction coefficients for very light initial contact forces and undoubtedly ensure that young shoots remain in place and are braced against supporting branches during the initial contact and twining process. Following twining, recurved spines act as a highly efficient anti-slip system ensuring that the twining, climbing stem does not slip down. The study found that micro spine climbing was particularly well-developed in species living in heterogeneous 3-D environments [24]. In this example, plant species have evolved a two-phase or two-step attachment strategy. As such, attachment does not rely on a single mechanism that will only work if the substrate type is the right one and if environmental conditions are ideal.

Other climbing plants can show series of physical, chemical, and structural “steps” that together optimise the final anchorage of the climbing plant to variable supports. In the English ivy [25], the plant develops up to four steps of initial contact of the growing root: form closure of the root surface with the substrate, chemical adhesion, and then further tightening to finally guaranteeing even tighter closure at a fine scale via root air reconfiguration, all contributing to the anchorage process. Like English ivy, the climbing cactus described here also climbs using adhesive adventitious roots that emerge from the stem. Unlike English ivy [25], the fine-scale actual adhesion process has not yet been studied in *Selenicereus*. In general terms the roots of ivy and *Selenicereus* function similarly in anchoring the climbing stems to supports. English ivy, however, lacks instantaneous hook attachment, and we suspect that this difference possibly limits the habitat and substrate diversity that ivy can climb on, such as on small branches in wind-prone environments.

### 1.2. Failure and Safety

The notion of attachment failure is also of great interest for attachment strategies in biology and for technical applications. Plants have evolved mechanisms to avoid failure, or when faced with unsustainable stresses, fail benignly, often in a way that is not catastrophic or lethal to the continued functioning. Climbing plants, particularly woody tropical lianas, are particularly well known for their flexible, “unbreakable” stems that can survive huge bending and twisting movements that would kill the stems and branches of “regular trees” [26,27,28]. They have evolved in this way in order to survive tree and branch falls and other disturbances in their life histories as slender stems in physically dangerous environments. We are also beginning to recognise that the attachment mechanisms of climbing plants have also evolved safety mechanisms that protect the plant from catastrophic failure [16]. In other words, an attachment system might partially fail if it has to, in the face of overwhelming forces, but it will not completely fail. Other attachment systems such as the micro-spine bearing stems of *Galium* are known to have extraordinarily extendable stems bearing attachment spines that require considerable elongation before failing [29]. In other plants such as the adhesive pads of the Virginia creeper, branched attachment systems respond to tensile forces by failing sequentially and thus reducing and escaping catastrophic strains but also remaining attached by remaining intact adhesive pads [30].

In climbing palms, hook-bearing rachises show simple ratchet-like, fail-safe mechanisms [31,32]. Modified leaves and fertile axes bear spines that attach to surrounding branches. They rely on a ratchet mechanism to maintain a guy-rope-like tension with the climbing palm and the trees. If tension in one of the connections becomes too tight during wind movement, the rows of spines along the palm are geometrically organised so that one or more spines might fail and shear away from the stem benignly. This loosens the tension and allows stronger hooks below to maintain tensile strain and thus “save” the positioning of the climbing plant between its supports [32]. In this example, the presence of pre-formed rigid hooks of variable resistance provides a safety-latch-like system and is entirely “passive”, not involving any adaptive growth. In the climbing plant world, mechanisms of attachment and fail-safe have to operate in fast and often violent environmental conditions, especially under high, gusting winds. Such conditions would otherwise strip the climbing palms of their hooks and risk irreversible and lethal falls into the darkness below the forest canopy.

### 1.3. Climbing Cactus Model

In this paper, we explore the two-step attachment system in the climbing cactus *Selenicereus setaceus* (Cactaceae) from dry coastal forests of eastern Brazil [8,22]. Cacti have highly adapted stems for water storage and photosynthesis, and this species shows a number of traits related to this condition: lightness of construction in terms of biomass, integration of a hydrogel-like material in soft tissues, and a simple layered organisation that has been the basis for new bio-inspired multi materials actuators [7,9]. Previous studies on the bio-inspired potential of the species identified that growth sequences of development (from young to old stages) in the stem can represent potentially adaptable blueprints for technical developmental models.

In this paper, we investigate the environmental significance, developmental processes, biomechanics, and failure mitigation mechanisms of the “hook and root”, two-step attachment strategy [22] (Figure 1). We aim to provide further potential bio-inspired openings for new practical and theoretical aspects for attachment in variable and unpredictable, changeable environments.

### 1.4. Questions and Hypotheses

(1) How are hooks and adhesive roots deployed by the plant in its natural environment? (2) What is the maximal strength of hooks and roots in terms of the connection with the soft body of the cactus? (3) Do attachment structures involve safety features that mitigate against catastrophic or dangerous failure? This is particularly important in terms of the light construction and the role of the stems acting as a water reservoir in a water-constrained environment. (4) How are mechanically rigid and strong materials (root and spine tissue) structurally integrated into a soft body? What is the tissue organisation allowing these composite organisations and what can we learn from them for permitting safe attachment and benign failure? (5) What aspects of two-step attachment and soft body life history provide insights for new technologies, particularly soft robotics, and materials sciences, involving light and soft body organisations?

## 2. Materials and Methods

### 2.1. Collection and Environment

Stems of *Selenicereus setaceus* (Cactaceae) [22] were observed in situ and collected for mechanical tests from the “Restinga” coastal lowland dry forests near the town of Buzios (22°44′49″ S, 41°52′54″ W, approximately 170 km east of Rio de Janeiro city), Rio de Janeiro State, Brazil. *S. setaceus* is a common component of the vegetation here where the climate is typically warm, dry, and with nearly constant light or strong winds. Mechanical tests were carried out on individual spines and roots collected from natural habitats. Healthy segments of cactus stem up to a metre in length with star-shaped to triangular cross-sectional stem profiles were pruned from healthy individuals in the field with sharp secateurs, wrapped in moist tissue paper, placed in plastic bags, kept in cool containers, and transported to the laboratory. Fourteen stems were used to test spine strength, from a total of 117 spines tested; fourteen stems (four of which were also used to test the spines) were used to test the root strength, from a total of 75 roots tested (Table 1).

### 2.2. Spine Tests

Mechanical tests termed hook strength tests (Figure 2A), following the terminology of [31], were carried out on individual hooks of the cactus to measure the maximum force (N) of attachment of one spine from each spine-bearing cushion from the apex to the base of distal segments of cactus.

To do this, a spine-bearing segment of cactus stem was trimmed from the cactus with a razor and clamped firmly in the base of a mechanical measuring device (Instron In Spec 2200, Instron Corporation, Norwood, MA, USA set up for tensile tests. Stem segments were clamped in such a way that one of the basally pointing spines (normally the central one) pointed downwards within 0° to 5° of the long axis of the vertical cross-head in the upper part of the device. Previous analyses of hook attachment organs have emphasised the interest of measuring hook resistance at variable angles to determine their effectiveness in different directions [11,33]. Because of the practical and geometrical complexity of multi-spine-bearing cushions, in this study, we preferred to test only downward-pointing spines in the same direction normal to the axis orientation. In our view, this would still provide relevant results concerning the principal supporting strength of upward-climbing-orientated stems. Further multi-directional analyses would be an interesting feature to study for this species but more feasible under controlled growth conditions.

A 60 mm length of steel fishing line (0.58 mm diameter) with a nylon sheath was tied into a loop of approximately 35 × 6 mm, and the knot was stabilised with a drop of cyanoacrylate resin (superglue) to prevent slipping. A relatively large loop was necessary to manoeuvre the loop between other spines and projections of the areole (spine pad). The loop was placed around the basal part of the spine at the point of junction between the spine base and the edge of the areole (Figure 2A). A relatively broad 0.7 mm nylon-coated fishing line was chosen, because following pilot tests, the narrower or uncoated line tended to cut into the spine base and cause a kind of failure that was not representative of attachment to natural supports. Since we measured only maximal strength in terms of load (N) at the spine bases, we judged that any slight compliance of the nylon sheath could be neglected in the mechanical pull tests. Attachment strength was measured by raising the steel loop against the spine at 0.2 mm per second with a force transducer cell of 125 N until failure when the crosshead was stopped. Failure was deemed to have occurred following a significant drop in resistance and/or a visible and significant deformation of the spine and areole organisation.

### 2.3. Root Tests

Mechanical tests referred to as root strength tests (Figure 2B) were carried out on individual roots of the cactus to determine the maximum strength of attachment of the extended root with its internal connection in the stem. Initial pilot studies of root attachment aimed to determine the attachment strength of roots to various substrates in the natural habitats via pull tests and/or peel tests in situ [30]. However, the geometrical variability (i.e., length of root development) and range of substrates was so variable in real-world conditions that this aspect of the plant anchorage could not be carried out without carefully controlled experiments of plants growing on substrates under controlled conditions. Instead, we measured the maximal strength of the root and its connection with the parent stem and its kinds of failure. From an anchorage perspective, this is of significant biological and functional interest, particularly in terms of how a “strong” tensile attachment structure is mechanically attached to a “soft” body.

In root strength tests, segments of cactus stem 10 to 30 mm in length, bearing attached roots, were cut away from the cactus stem with a razor. A simple customised “L-shaped” platform constructed from flat steel plates with a central aperture was connected to the base of the Instron measuring device. The flat surface of the cactus with an emerging root was placed on the underside of the horizontal steel plate, and the root was passed through a circular aperture of the steel plate and clamped to a tensile grip attached to the force transducer of the measuring device. The cross-head was then raised at 0.2 mm per second with a transducer force cell of 125 N so that the root was put under tension against the cactus stem until failure of the root-cactus stem connection and the root base and root trace had been pulled out of the cactus stem.

### 2.4. Functional Anatomy

Transverse and longitudinal sections of spines and roots and their connective tissues with the cactus stem were carried out on specimens that had been tested mechanically, or that were adjacent to tested samples.

Samples including different stages of development of spines and roots were first trimmed to pieces of tissue up to 1 cm in thickness. Samples were then embedded in paraffin (Leica Surgipath Paraplast) using an automatic embedding station (Microm STP120). Specimens went through a dehydration series of ethanol (70%, 90%, 95%, and two times 100%) followed by D-limonene and then in liquid paraffin at the end of the cycle. Samples were then embedded using an embedding station (Microm EC350-2) after being manually positioned into the desired sectioning orientation. After cooling/polymerisation of the paraffin, each paraffin block was cut with a rotary microtome (Leica RM2245 Rotary Microtome) to thicknesses of 9 to 10 µm.

Blocks containing the cactus stem were sectioned immediately before mucilage began to exude from the stem. If a sectioned stem sample has exuded too much mucilage, the paraffin block was melted, re-embedded, and sectioned again. Following initial tests with this species and these protocols, we elected to not remove the mucilage from the biological material since sections were not overly affected by mucilage and because mucilage is likely an integral aspect potentially influencing cell organisation and cell shape.

Blocks containing a cactus root were put into cold water to soften the root tissues before cutting 8 μm thick sections. Series of sections were made from the base to the middle of the root. Sections were mounted in warm water (40 °C) on glass slides previously covered with albumin then stained with 1% Safranin and 0.5% Alcian blue through a staining automat (Leica Autostainer XL, ST5010). Slides were then mounted with a coverslip using synthetic resin (Cellpath DPX (Phthalate Free) Mounting Medium). Cactus sections were observed, and images were captured using a digital microscope (KEYENCE VHX 7000).

### 2.5. Data Handling and Statistics

Data were captured during mechanical tests and stored using an electronic interface and Instron S9 software and then transferred to data frames. Comparisons of mechanical properties and morphology were tested via non-parametric Kruskal–Wallis tests, followed by Mann–Whitney post hoc tests carried out in the software PAST [34]. As in other studies of climbing plants, there was a difficulty of obtaining unequivocally separate individual plants from what are complex, highly branched, clonal life forms while sampling from the same habitats and geographical locations [24]. We therefore elected to apply non-parametric tests for comparing biomechanical values of median values and post hoc tests as well as non-parametric Spearman rank-order correlation coefficients.

## 3. Results

### 3.1. Morphology, and Deployment of Spines and Roots

The cactus has a climbing habit and attaches to branches and trunks of shrubs and trees, rocks, and soil surfaces with spines and adhesive roots (Figure 3A–E). Spines occur in groups of two to five (usually 4–5) on pad-like extensions (areoles) at regular distances (9 to 84 mm) along the ridges of the triangular cross-section of the cactus stem (Figure 3C,E and Figure 4A–D). When mature, spines are 2.9 to 0.71 mm in length (Table 1) and generally recurved with points directed downwards or to the sides of the stem and resemble grappling hooks (Figure 4B–D). Spine size and alignment changes during development from apically pointed near the apex to basally pointed below (Figure 4A–D). The youngest spines at the apex are enveloped by the areole (Figure 4A). Downward-pointed spines anchor the stem to supports such as tree trunks, rocky surfaces, and concrete posts (Figure 3B,C). Spines also anchor stems in shrubby branched, windswept vegetation (Figure 3D–E) (Appendix A).

**Table 1 biomimetics-08-00220-t001:** DIMENSIONS OF STEMS, ROOTS AND SPINES. All means of length, diameter and stem fresh mass are followed by the standard deviation and minimum and maximum values between brackets. n = number of samples; na = not applicable.

Stems	Length (cm)	Diameter (mm)	Fresh Mass (g)
n = 25	61.58 ± 27.02 (11.50–131.50)	13.32 ± 3.85 (6.70–21.28)	66.72 ± 35.44 (5.10–175.10)
Roots	Length (cm)	Diameter (mm)	Fresh mass (g)
n = 75	na	1.10 ± 0.28 (0.30–1.73)	na
Spines	Length (mm)	Diameter (mm)	Fresh mass (g)
n = 38	2.09 ± 0.47 (0.71–2.97)	na	na

Mature spines are rigid, tapering, cylindrical structures. They are formed by elongated cells with thick lignified secondary walls surrounded by layers of cells with thicker cell walls (Figure 4E,F arrows). The lignified tissue of the spine base is continuous with an extended mound of soft unlignified tissue of the stem (Figure 4E,G) of heterogeneous unlignified cells; serial sections demonstrated that the spine base is concave and is seated on top of the non-lignified mound tissue below (Figure 4E). There is a sharp transition between spine base tissue and mound tissue beneath (Figure 4E, arrows).

Roots appear on all surfaces of stems (Figure 5A,B) and are initiated as small bumps on the surface of the stem (Figure 5C,D). These arise from xylem traces of the wood cylinder and grow through the soft tissue of the cactus stem (Figure 5E) displacing and compressing adjacent tissue (Figure 6A,B). As the developing root tip emerges from the cactus stem wound repair cells of the cortex seal off the spaces formed around root where it emerges from the surface (Figure 6B, arrows). Following the emergence and sealing, roots extend from the cactus stem and rapidly develop tufts of fine root hairs (Figure 5D). Although not measured, our observations suggest that young, emerged roots show directional growth (possibly tropisms) towards substrates, triggered by the presence of water or humidity. When in contact with a substrate and after adhesion of root hairs, root growth continues along the surface of tree bark or stone or concrete surfaces (Figure 5A,B).

### 3.2. Spine Strength

The spines tested in tensile strength tests presented three types of failure during the mechanical tests: (i) elastic and non-elastic deflection, where deflection of the spine at the spine base resulted in a large change of angle resulting in the steel loop slipping from the spine; (ii) rupture of single spine base with the subtending areole, where the spine base of the tested spine detached cleanly from the softer tissue of the areole below (Figure 7A); (iii) rupture of the entire basal spine cluster and the areole base, where the entire group of spine bases detached from the softer subtending spine bases (Figure 7B).

The most frequent mode of failure was at single spine bases, representing 64% of all spines tested. This was followed by failure via elastic and non-elastic deflection (26%). A third mode of failure involved separation of the entire cluster base (less than 10%).

The highest loads were observed prior to failure of the entire spine cluster but resulted in the loss of all attachment capability (mean maximum loading forces 5.72 ± 3.27 N; Table 2). Evidence of this kind of failure was also noted in natural habitats where entire clusters had been lost but were completely sealed by periderm or wound tissue resembling that formed around the emerging root (Figure 7C).

The lowest loads were observed when spines deflected elastically or non-elastically but not separating from the pad (mean maximum loading forces 1.21 ± 0.95 N; Table 2).

Intermediate loading forces were observed for the single spine base failure mode (mean maximum loading forces 2.80 ± 1.59 N; Table 2) (Figure 8A). On the basis of the overall mean of single spine strength, one single spine would be theoretically capable of supporting on average 285.5 g of stem mass, which corresponds to a stem length of 2.62 m.

There is a weak-to-moderate correlation between maximum loading force and the distance of tested spines from the stem apex (Rs = 0.44; *p* < 0.001) (Figure 8B). This suggests that in young spines, failure is caused by lower loads and spines remain fully or partially functional afterwards. Contrary to expectation, there was no correlation between distance from the apex and elastic/non-elastic type of failure nor spine cluster failure. Elastic and non-elastic failure appears to be more frequent in younger stem stages while entire cluster failure appears to occur randomly. The single spine base failure showed only a weak correlation between loading forces and distance from the apex (Rs = 0.35; *p* < 0.01), suggesting that spines tend to be less strongly attached to the pads near the apex and that the transition of the spine base towards the soft internal tissues is more compliant at in younger stages nearer the apex.

### 3.3. Root Strength

Roots under tension resisted the tensile force until failure of the root or part of the root either within the cactus stem or between the cactus stem and the grips (Figure 9A–F). Notably, root failure never caused tearing or damage of the stem besides a neat aperture (Figure 9A). Four categories of root-stem failure were observed:Transverse fracture of the root external to the cactus stem; this involved simple transverse fracture of the root outside of the cactus stem but not at the position of the clamps gripping the root during the test.Transverse fracture of the root trace or root base within the soft cortex of the cactus stem (internal); this involved simple transverse failure of the root inside the stem but without any sheared flanges of tissue from the wood cylinder (Figure 9B).“T-shaped” fracture at the wood cylinder (Figure 9C,D); this involved separation of the root from the internal wood cylinder with flanges of sheared xylem tissue derived from the xylem cylinder above and below the point of departure of the original root trace. During the test, the flanges of sheared xylem tissue are dragged through the stem and exit the stem with the rest of the root.“L-shaped” fracture at the wood cylinder; this was similar to the T-shaped fracture but only involved one flange of sheared tissue from the main wood cylinder (Figure 9E–G).

“T-shaped” and “L-shaped” failures were the most frequent types observed, representing 44% and 21%, respectively, of the total of 75 roots tested. Internal and external transverse fractures represented 19% and 16% of the total. “T-shaped” and “L-shaped” failures showed the highest loading forces and were not statistically different (Figure 10A) (mean force values attained for “T- and L-shaped” failures were, respectively, 15.20 ± 9.71 N and 17.57 ± 10.44 N; Table 2).

**Table 2 biomimetics-08-00220-t002:** TYPES OF SPINE AND ROOT FAILURE: Mean values of the maximum loads for (1) The three modes of spine failure and (2) The four modes of root failure. All means are followed by the standard deviation and minimum and maximum values in brackets. n = number of samples.

(1) Spines: types of failure	Single spine base failure (n = 75)		Entire spine base failure (n = 11)		Elastic and non-elastic deflection (n = 28)	
Max Load (N)	2.80 ± 1.59 (0.18–6.75)		5.72 ± 3.27 (1.84–11.92)		1.21 ± 0.94 (0.06–3.76)	
(2) Roots: types of failure	T-shaped (n = 33)	L-shaped (n = 16)		Transverse fracture (internal) (n = 14)		Transverse fracture (external) (n = 12)
Max load (N)	15.20 ± 9.71 (3.79–39.69)	17.57 ± 10.44 (5.93–36.05)		9.05 ± 6.08 (2.20–21.44)		9.90 ± 5.71 (2.57–19.57)

The transverse failures (internal and external) did not apparently require such high forces (means 9.05 ± 6.08 N and 9.90 ± 5.71 N, respectively; Table 2) to rupture the anchorage of the root to the central cylinder or the structure of the root itself, such as T- and L-shaped (Figure 10A). The internal transverse fracture type was observed to be more frequent in younger stem segments, where the inner rigid core of xylem is not so developed (observations made during the tests). The statistical test comparing medians showed that the external transverse failures showed little or no statistical difference with the more resistant internal T- and L-shaped failures (Figure 10A). On the basis of overall mean root strength, one single root would be theoretically capable of supporting an average of 1398 g of stem mass, which corresponds to a stem of 12.91 m of length.

There was a strong rank correlation between maximum loading force and root diameter for all failure modes considered together (Rs = 0.77, *p* < 0.05) (Figure 10B). Viewed as independent categories, there was a strong correlation between maximum load and root diameter for all categories of failure: T-shaped (Rs = 0.76, *p* < 0.05) and L-shaped fractures (Rs = 0.80, *p* < 0.05), transverse fracture (internal) (Rs = 0.85, *p* < 0.01), and transverse fracture internal (Rs = 0.70, *p* < 0.05), indicating that larger diameter roots are stronger. Further analyses more closely linked to developmental stage, and anatomical organisation might further explain why external transverse failures were less well linked to root diameter. 

## 4. Discussion

### 4.1. Failure without Tears

Climbing is risky for real plants as well as technical artefacts. Climbing plants have evolved many mechanisms to ensure that attachment functions in the environmental conditions for which they have adapted to. For example, climbing palms have developed many mechanisms for climbing in the absence of wood [31,32,35]. Many woody climbers have evolved diverse strategies of attachment that are based on specialised wood growth [16]. In this study, we focused on a climbing cactus and the mechanisms of attachment in plant body highly modified for water storage and photosynthesis. This requires integration of tough, stiff resistant materials with a “soft” water-storing body. Previous studies have underlined the “soft” construction of this species [8] and how it can form the basis for developmental, bio-inspired stem movements that respond to changes in the environment [7,9]. This current study has shown that despite a soft body organisation, the cactus is capable of producing two distinct and complementary attachment mechanisms. Both are strong enough to support significant lengths of climbing stem. Furthermore, in perturbed conditions such as strong winds, both attachment systems fail benignly and do not lead to catastrophic tears of fissures that would jeopardise the integrity and survival of the stem.

### 4.2. Spine Deployment

Deployment of pre-formed spines shows a characteristic pattern of development. Young spines are pointed apically and then develop both downwards and in multi-directions. Some plants are known to use growth processes to “push through” and navigate past surrounding plants [36]. Apical growth would be hindered if anchoring spines were pointing forward rather than backwards, and this is reflected in the rapid backwards-pointing geometry observed below the apex of the climbing cactus. When searching stems are in bushy, densely branched environments (Figure 3), stems undoubtedly “push-through” branch networks, and spines are passively deployed to lodge the stems into position. Rapid deployment of recurved spines below the apex undoubtedly assists vertical climbing (Figure 3). Sharp spines penetrate relatively soft bark tissue of trees and undoubtedly brace the vertically orientated stem to permit step two connection by root growth.

Finally, recent studies have suggested that positioning recurved spines on stems with triangular rather than circular cross-sections are particularly efficient at forming attachments in highly unstructured three-dimensional situations [24]. Spine deployment is likely the principal mode of anchorage in the cluttered, shrubby, and tightly branched environments of the wind-swept coast (Appendix A). In quieter understorey situations, it also represents a steadying or stabilising effect, increasing the likelihood of successful root deployment on tree trunks, rocks, and ground surfaces (Figure 3).

### 4.3. Root Deployment

Roots are deployed very differently compared to spines. They are not preformed or arranged in spirals along the stem. Instead, anchoring roots develop adaptively, possibly in response to changes in light or humidity, as well as proximity of supports. A key difference is that root deployment is not instantaneous but involves growth processes, movements, and likely tropisms towards and onto substrates. Root development is therefore slow, relative to the deployment of spines and also requires that the cactus stem is braced or held steady to the support. Our observations indicate that roots are generally deployed on large, broad, or fixed supports such as tree trunks, rocks, and the ground surface rather than small branches.

Spine and root deployment in the same plant reinforces recent ideas that climbing plants can combine different modes of attachment and optimise attachment under varied environmental conditions [24]. The cactus shows deployment strategies for both moving and still environments. The combination of these (i) fast-passive and (ii) slow-adaptive strategies using is a promising strategy for soft robotic artefacts that are designed to anchor safely in widely different environmental conditions and on different substrates.

### 4.4. Spine Failure Strategies

Three modes of spine failure are linked to the deployment strategies of the cactus. The first, involves forward-pointing, young spines provide the least resistance to apical growth. Their failure is linked to elastic deformation. Young, apical spines are partially shielded by the edges of the areole, preventing young spines catching on surrounding branches and impeding forward growth.

Second, the most frequent kind of spine failure occurs around the base of the single spine. Failure occurs via propagation of a crack between the mound of soft tissue and the concave surface of the lignified tissue. It represents a benign failure where (i) soft tissue beneath the spine remains intact, thus limiting damage and water loss, and (ii) only one spine is lost from the pad. The loss of one spine can be viewed as a single, “pop-off” mechanism, limiting damage to the plant and ensuring functionality of remaining spines. The mechanism is similar to failure characteristics in *Rosa arvensis* [33], where damage to the plant body is also limited. There are similarities in the overall organisation of thin-to-thick-walled fibre cells of the spine organisation in the cactus and *Rosa*. Multi-spine areoles of the cactus differ from the single hook organisation in *Rosa*. It is important to note that cactus spines are highly modified and reduced leaves [37], whereas prickles of *Rosa* are outgrowths that originate from tissues below the epidermis [38]. Similar kinds of anchorage and benign failure have evolved in phylogenetically disparate groups in roses and cacti. Further studies should investigate how the angles of the applied force on individual spines [33] and spines of the same pad influence attachment force and spine-pad integrity.

The third mode of failure resulted in the loss of the entire basal pad and all spines. This was less frequent than other failure modes but still showed clean separation of the soft tissue mound and the spine-bearing areole. Field observations (Figure 7C) showed evidence of both single and entire pad loss under natural conditions as well as periderm-like wound tissue sealing off areas exposed by pad loss. Although all spines are lost, wound closure takes place following pad loss.

On the basis of average mass and the length of searching and climbing stems, a single spine could theoretically support up to 285 g of climbing stem equivalent to approximately 2.6 m length of stem. This means that only a tiny amount of biomass expended on a spine can return significant mechanical support at a whole plant scale. We emphasised above that spines can deploy and fix a cactus stem in moving, wind-prone environments. Gusting and branch movements can no doubt initiate rapid, violent, and unpredictable mechanical stresses. It would seem that a multi-level range of spine attachment strength including (i) elastic deflection, (ii) single spine (“pop-off”), and (iii) high force-multi-spine loss offers different levels of safety for survival under different magnitudes of force. None of the failure modes appear to involve tearing or opening of the cactus water-storage system. This is consistent with ductile kinds of failure observed for stems failing under self-loading in this cactus species [8].

### 4.5. Root Failure Strategies

Root deployment and failure show a range of strategies linked to the water-conserving soft body life history of the cactus. Both L- and T-shaped failure of the roots fail at similar high forces. On the basis of the mean strength of all root failure categories, a single root could theoretically support a stem mass of 14 kg and a length of approximately 12.9 m, providing that the force of attachment between the root and the substrate would sustain this. Though it was not possible to test them meaningfully, our observations and initial pull tests of roots attached to tree trunks would likely sustain this magnitude of force (Figure 3B).

High resistance of L- and T-shaped failures is ensured by the attachment of the lignified tissues of the root trace to the lignified tissue of the wood cylinder (Figure 5E and Figure 9C,E). Failure involves tangential separation of the wood fibre elements around the point of departure of the root xylem trace.

Resistance against failure is likely established by thick-walled, mature fibre elements of the wood cylinder and root xylem. Tensile forces act tangentially against the stem wood cylinder, and this orientation is far weaker that forces applied along the grain (longitudinally) of the wood cylinder [39]. The images of failed L- and T-shaped failures indicate that shearing causes tear-like flanges of xylem tissue that exited the plant stem at the end of the test.

Transverse failures of the root system did not reach the higher resistance values (25–40 N) of L- and T-shaped failures. The plot of root diameter and maximum load shows that higher values only really occur in root diameters greater than around 1 mm (Figure 10B) where we see mostly L- and T-shapes. Higher root strengths likely require higher degrees of wood cylinder growth, cell wall thickness, and lignification. The data suggest that before this point, younger roots risk failing transversely at lower forces. Further studies should establish whether root development and strength only increase if the root contacts a support. One possible explanation for the heterogeneity of the transverse fractures is that roots appear to die off and dry under natural conditions if not attached to a substrate. This is difficult to control for unless the study is undertaken under controlled conditions.

### 4.6. Conclusions: Potential Technical Concepts and Technical Applications

This paper has demonstrated that the presence of a two-step attachment mechanism in a climbing plant is a great asset for its biology and survival. Here, we discuss how some of these functional advantages may serve as conceptual or technical inspiration for new technologies, particularly artefacts based on climbing plants and other soft robotic applications.

First, the two-step attachment strategy allows a climbing plant to attach to widely different substrates and environments; in this case, it is clear that spines definitely permit anchorage in the moving environment of narrow branches of wind-swept coastal environments (Appendix A). Secondly, the second step, root anchorage, permits stronger, solidly fixed, and irreversible anchorage to solid and stationary substrates. This dual kind of attachment strategy would be especially appropriate for robotic artefacts that need to explore variable and unpredictable environments [3]. The two attachment mechanisms are also integrated and work together. In some cases, spines can steady or brace the climbing stem in position while adventitious root growth can develop while the stem is held in place. For technical artefacts dual anchorage mechanisms are arguably better than one. They can expand the potential substrates and work together to optimise attachment.

The two-attachment mechanisms also operate on different time scales; pre-formed, ready-to-deploy, recurved spines can form an attachment instantaneously with supports in moving environments and situations. In the real world of climbing plants, these can fix passively to surrounding supports and can also act as safety mechanisms and catch on to supports in the event of falls. Root growth, on its own, would have little chance to attach to moving supports and would not catch onto things in the event of a fall. Artefacts could incorporate anchorage structures that attach instantaneously as well as structures that require growth towards specific targets and objects. For technical artefacts, anchorage systems could also be more specifically designed to undergo slower additive growth towards connecting with a more targeted object or surface.

Our study identified many features that can be interpreted as benign failure. Research into new materials and structures for spine-like attachment and anchorage in technical artefacts is gathering pace [10,11]. Ongoing research is examining how natural hook systems are geometrically and materially designed to best function for anchorage [11]. The hook-like mechanisms we observed suggest that combining clusters of spines in readily pre-fracture designed pads have what we could call a “save” and “sacrifice” mechanism.

The ability of plants to produce root-like attachment organs adventitiously (from parts of the plant that are not roots) is a highly adaptive aspect of a plant’s functional biology. It means that attachment organs are not just produced at the growing apex but can also be deployed adaptively at other (older) stages of development from the plant body. It provides an opportunistic means of providing attachment points where they are needed, in terms of the overall plant body—it is never too late to produce an attachment point.

Soft robotic artefacts capable of axial additive growth of various kinds are now a technical reality [1,4,6]. Combining additive growth of an artefact with anchoring capabilities is a technical challenge. However, the adventitious growth, emergence, and deployment of adventitious roots in the cactus species presents interesting biological features that might offer some clues of how such systems could operate in technical projects.

The deployment of the cactus root system detailed above shows similarities with recent bio-inspired growth via extrusion technologies [40]. Our micrographs indicated that root growth pushed through the soft tissues of the cactus and hydrogel. The root tips then reached the stiffer tissues of the stem surface as “bumps” where the root cap is in close proximity to the cactus skin. Root deployment necessitated rupture of the cactus skin, but this occurred benignly without tearing or exposure of the hydrated cactus tissue to the outside. Indeed, spaces around the emerged root were sealed off by localised periderm development of the cactus stem.

The root strength tests indicated that root failures normally occurred deep inside the hydrated cactus body and that failure ended with a neat perforation wound at the stem surface without tearing. Tissue repair following mechanical damage and security against dehydration from wounding are becoming more fully understood in succulent plants that have light organisations and high requirements for water storage [41]. The deployment of the cactus root system described here involves a kind of benign failure and repair when the cactus stem is punctured by growth of the root from the inside. Cellular scale adaptations using additive growth or extrusive technologies might be important future requirements to ensure that soft robotic systems remain closed and sealed against the outside environment, thus enabling artificial adventitious growth from soft-bodied artefacts.

Finally, the construction of technical artefacts based on hard-soft biological materials such as in the cactus stem [7,9] needed careful consideration and testing of the interface characteristics and strength of contact between adjacent hard and soft tissues. Such interfaces permitted adaptive bending-like movements in response to humid and dry conditions. The benign failure of cactus spines indicated that the concave–convex interface between soft areole mound and hard lignified tissue of the spine base could readily separate without tearing and further wounding. The mechanism represents a kind of safety mechanism that depends on a strong but also separable interface between different layers of tissue. New research for optimised multi-materials, especially those based on hard and soft combinations, should look further at the systems present in nature and especially in plants. There is much potential for understanding how natural systems ensure that tissues normally never separate, but in some situations, they are adapted to do so safely.

Lastly, the anchorage mechanisms discussed in this paper have resulted from evolutionary and developmental processes that have shaped and modified pre-existing attributes of pre-existing plants. For example, the attachment spines in cacti represent previously defensive structures against herbivores, and prior to that, they were leaves. Evolutionary biologists refer to such changes as exaptations [42], where a structure’s function changes during evolution. The anchorage capabilities of the spines (modified leaves) and roots (adventitious development from stems) both likely represent exaptive changes of functions in cacti. Recent technological models based on this cactus have explored the possibility of using variable developmental pathways for developing functionally variable artefacts [7,9]. We propose that exaptive patterns of transition are instructive for constructing technical systems that are readily modifiable, using similar material components but organised differently to perform quite different tasks.

## Figures and Tables

**Figure 1 biomimetics-08-00220-f001:**
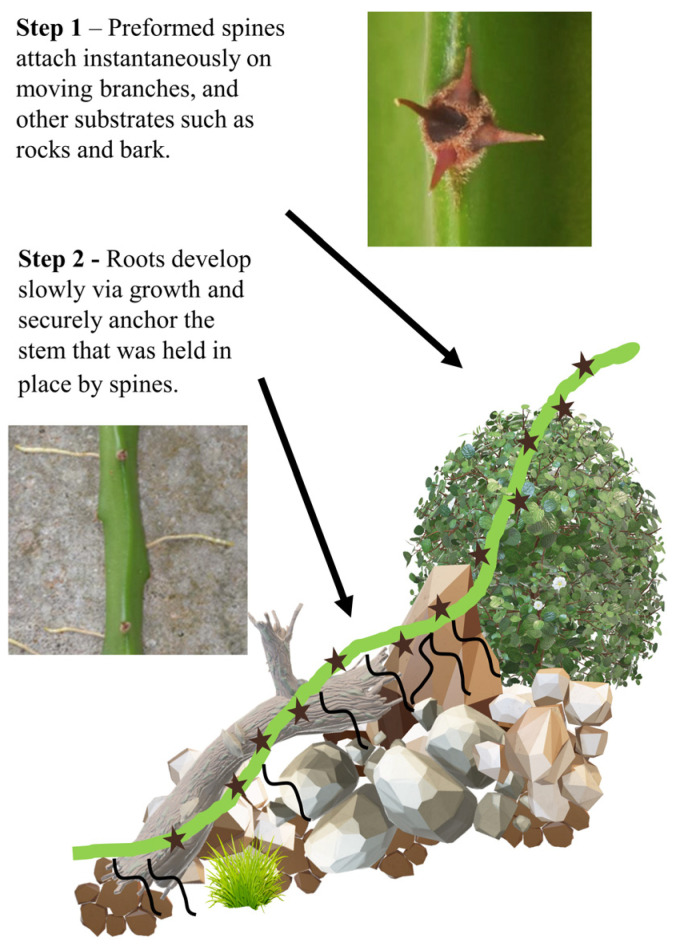
Overview of two-step attachment (modified from Soffiatti and Rowe (2020) [22]). In step 1, recurved hooks are pre-formed and attach instantaneously to many substrates, including soil, rocks, tree trunks, leaves, and branches in wind-prone environments. Hooks also provide bracing for the step 2 attachment by slow-growing roots that emerge from the stem and anchor the plant stem more firmly to many different kinds of substrate. In this paper, we assess the strength and functional significance of these two kinds of attachment and the potential technical applications for two-step attachment strategies.

**Figure 2 biomimetics-08-00220-f002:**
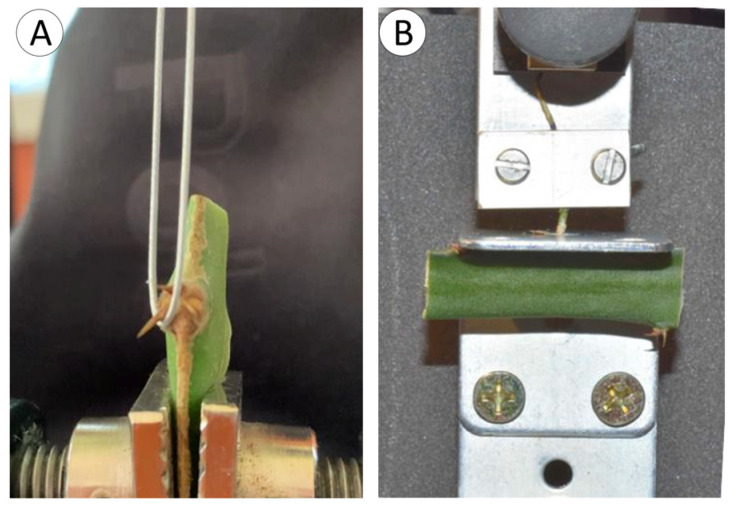
Mechanical test set ups used to test individual spine and root strength. (**A**) Hook strength test set up, using a steel fishing trace on a loop, placed at the base of a single spine. (**B**) Root strength test set up, using a horizontally steel plate with a hole through which a single root was pulled against the flat surface of the cactus stem.

**Figure 3 biomimetics-08-00220-f003:**
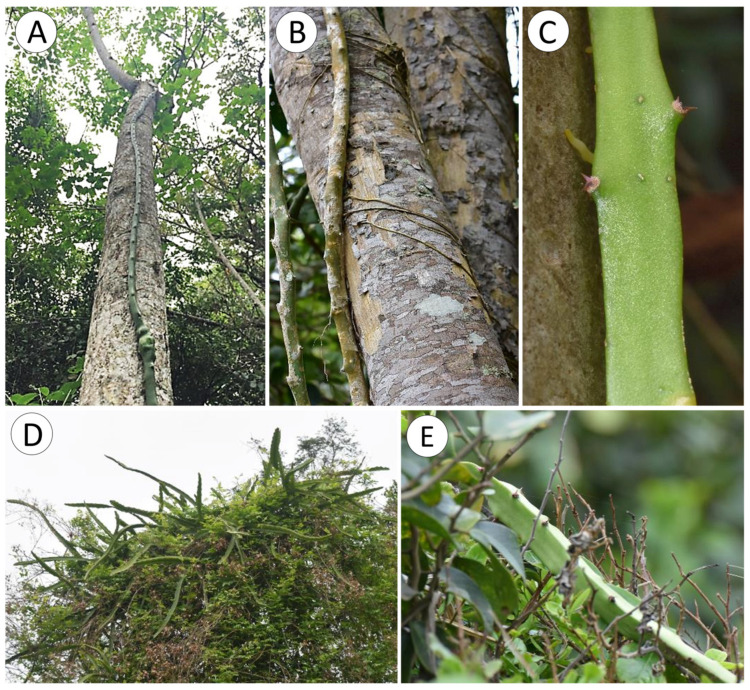
Selenicereus setaceus in its natural habitat in the dry coastal forest formations of Rio de Janeiro State, Brazil. (**A**) *S. setaceus* is a climbing cactus that often climbs along tree trunks using spines and roots. (**B**) Stem of *S. setaceus* firmly attached to a tree trunk via several roots. (**C**) Detail of stem showing spine clusters and the development of a young root. (**D**) Apical branches (searchers) reach the forest canopy in search of light. (**E**) Succulent, photosynthesising stems have to overcome several obstacles and grow across heterogeneous environment.

**Figure 4 biomimetics-08-00220-f004:**
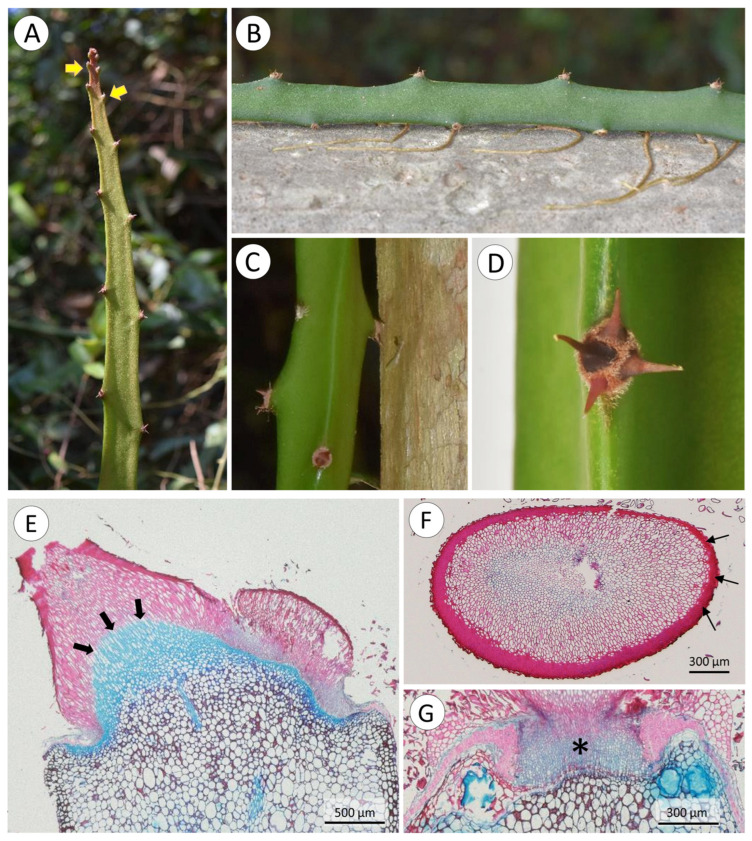
(**A**) Young branch of *S. setaceus* with spines located close to the apex pointing forward. As stem and spines mature, they change direction and point downwards. Note cushion of tissues around spine base near apex (arrow). (**B**) Stem attached to concrete, showing clusters of spines and roots. (**C**) Detail of spine cluster attached to a tree trunk. (**D**) Spine cluster, generally formed by 3–5 spines pointing in several directions. (**E**) Longitudinal section of a spine, showing a rigid structure composed of lignified cells and abrupt transition to soft tissue of the mound below (arrows). (**F**) Cross-section of a spine base, showing the external layers of very thick cell-walled cells (arrows). (**G**) Details of the transition between the base of the spine and the internal soft tissues (*).

**Figure 5 biomimetics-08-00220-f005:**
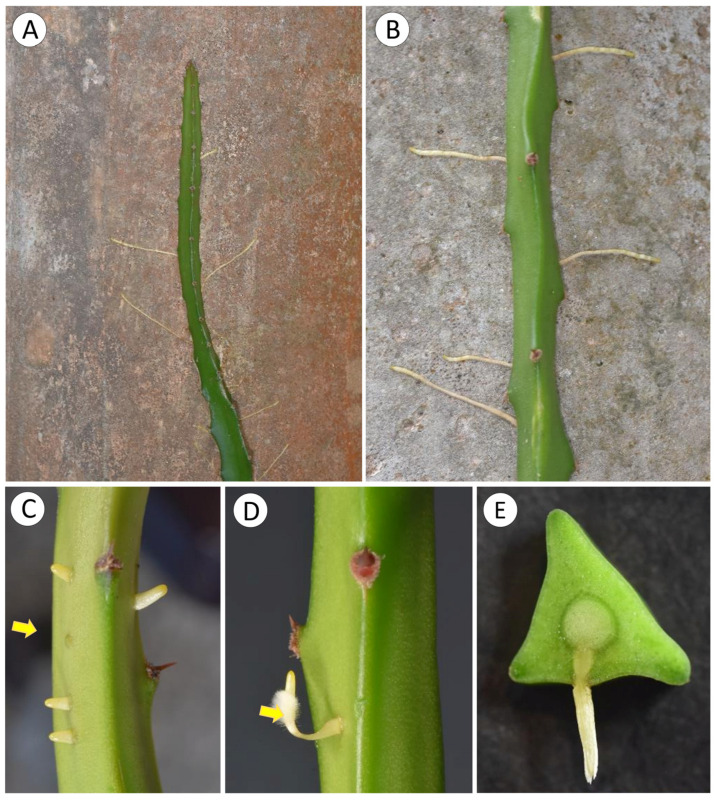
(**A**) Young stem forming several roots. (**B**) Mature portion of a stem with roots forming in several directions. (**C**) Initial phase of root formation with small bump (arrow) and emerged roots. (**D**) Young root with many root hairs (arrow). (**E**) Cross-section with root zone of root attachment to core of xylem and emergence point through skin.

**Figure 6 biomimetics-08-00220-f006:**
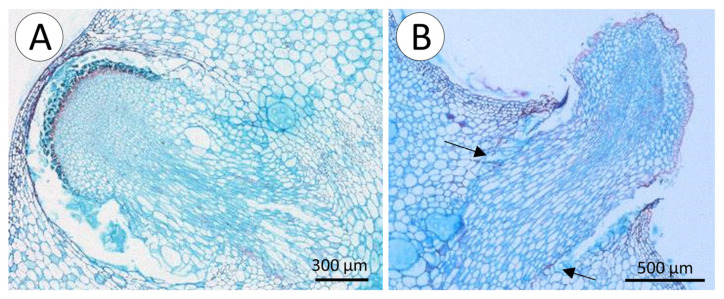
Cross-sections of root. (**A**) Young root just about to break through skin to continue its development outside stem. Internal soft tissues of the stem are pushed aside as root develops within the stem and collapse around the newly formed root. (**B**) Young root ruptures the skin and emerges from the stem. Note healing of surrounding tissues (arrows), without much damage to the internal structure of the stem.

**Figure 7 biomimetics-08-00220-f007:**
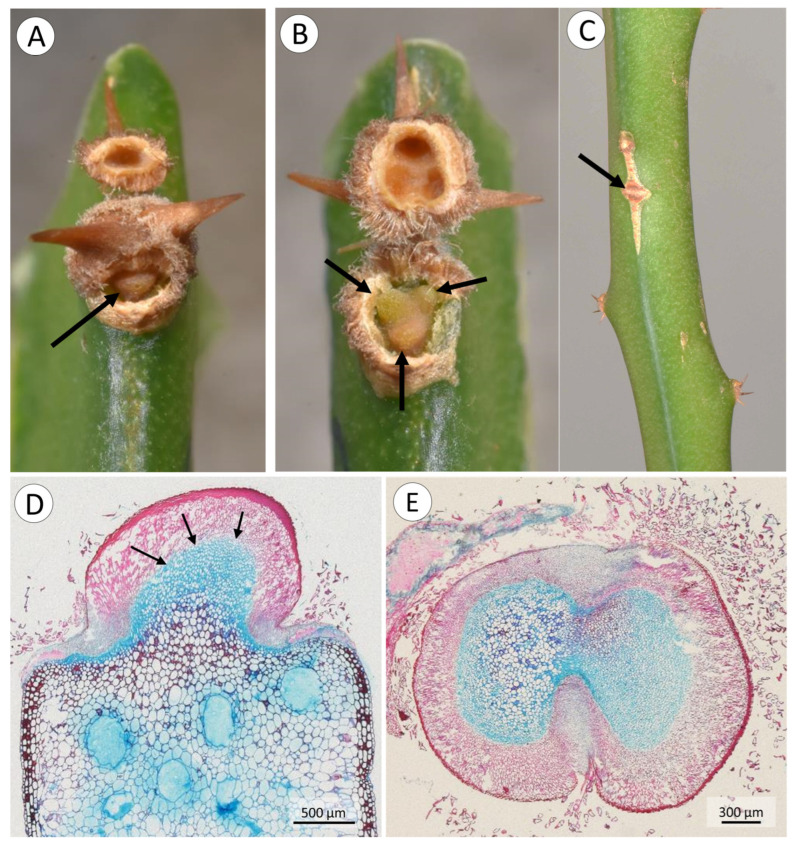
The two most common types of spine failure during mechanical tests. (**A**) Single spine base failure, where one spine is removed with its base from the pad. Note the undamaged internal mound within (arrow). (**B**) Entire spine base failure (whole pad). Note the three exposed and undamaged mounds, where each one of the three spines were positioned prior to the test. (**C**) A region of a spine cluster that healed naturally (arrow). (**D**) Longitudinal section of a single spine, showing the connection between the rigid spine base and the soft mound tissue (arrows). (**E**) Cross-section of the transition region between spine base (in red) and soft mound (blue).

**Figure 8 biomimetics-08-00220-f008:**
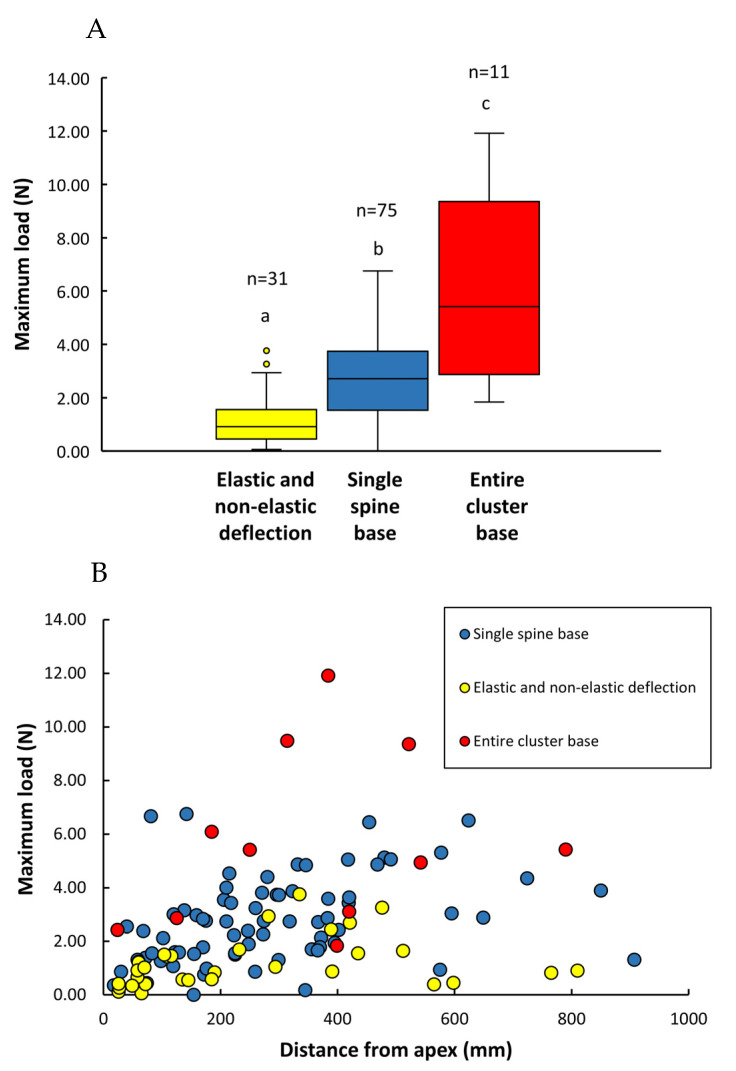
(**A**) Maximum loads obtained in spine strength tests for three types of spine failure (single spine base, elastic and non-elastic deflection, entire spine base; see Figure 6) in stems of *Selenicereus setaceus*. Highest loads are observed when the entire pad is separated during the test of a single spine, resulting in the loss of all attachment capability for that pad. Lowest values are observed when spines deflect elastically or non-elastically but do not separate from the pad. Inner lines: medians; boxes: 25th and 75th percentiles; whiskers: maximum and minimum values; letters indicate significant differences between median values of maximum loads between the types of spine failure (Kruskal–Wallis followed by Mann–Whitney post hoc test; *p* < 0.05). (**B**) Bivariate scatter plot of maximum load (N) and distance of individual tested spines from the stem apex (mm) according to the types of spines failure measured spine strength tests of stems of *Selenicereus setaceus*. There is only a weak-to-moderate correlation between maximum load and the distance from the apex. At younger stages of development, spines tend to be less strong, and the transition of the spine base towards the soft internal tissues is more compliant. Spearman’s correlation: all modes of failure (n = 117) Rs = 0.44 *p* < 0.001; single spine base (n = 75) Rs = 0.35 *p* < 0.01; there was no correlation between the other types of failure with distance from the apex.

**Figure 9 biomimetics-08-00220-f009:**
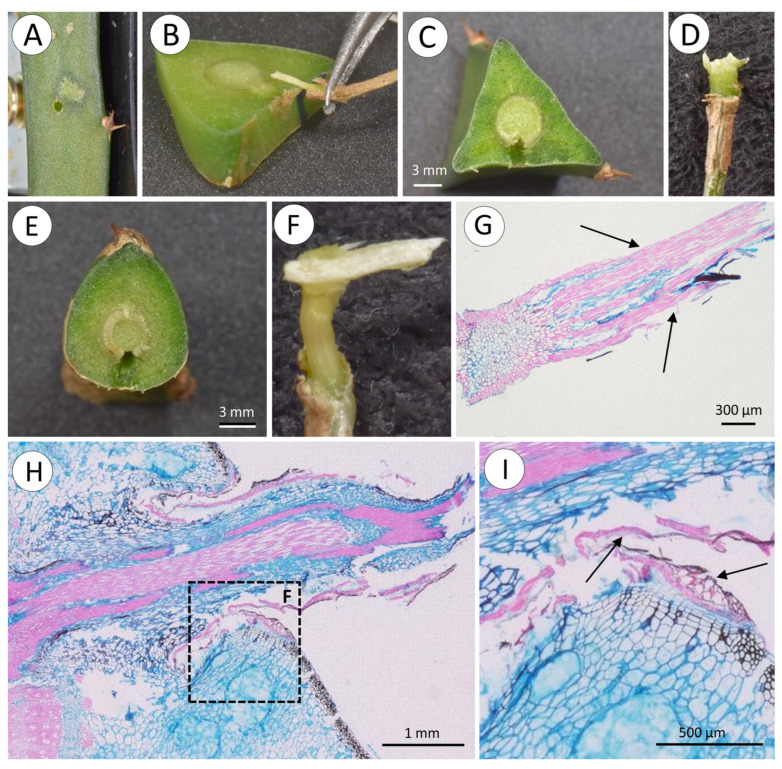
(**A**) Clean aperture left open after the removal of a root. (**B**) One of the four types of root failure: transverse fracture from the wood cylinder (internal). (**C**) Cross-section of a stem after a pull-out test with a T-shaped failure; the root ruptures part of the xylem cylinder structure where it was anchored and the outer soft tissues without much damage. (**D**) T-shaped fracture at the wood cylinder; the root retains two flanges of sheared xylem tissue. (**E**) Cross-section of a stem after a pull-out root test with an L-shape failure; the root ruptures part of the xylem cylinder structure where it was anchored and the outer soft tissues without much damage. (**F**) L-shape fracture at the wood cylinder; the root retains one flange of sheared xylem tissue. (**G**) Cross-section of the L-shape failure: note the amount of lignified xylem tissues (stained pink) (arrows) removed along with the root. (**H**) longitudinal section of a root emerging through the skin; the points where the root ruptures the skin are healed with the formation of a peridermis. (**I**) Detail showing the repair tissue (peridermis, arrows) formed at the points where the root broke the structure of the skin.

**Figure 10 biomimetics-08-00220-f010:**
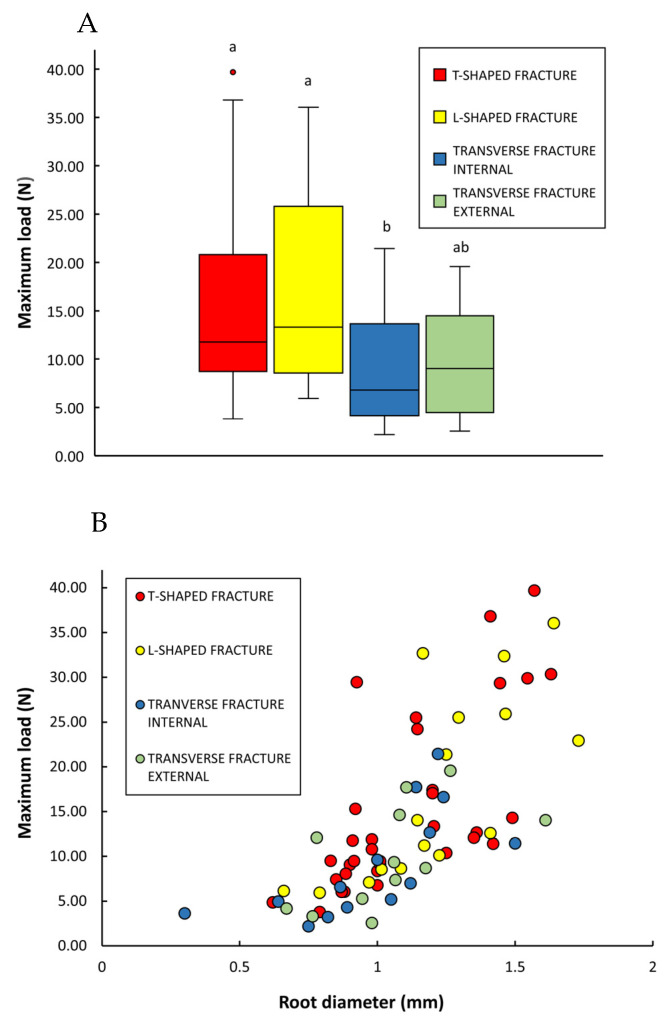
(**A**) Maximum loads obtained in root strength tests of *Selenicereus setaceus* for four types of root failure: T- and L-shaped fracture at the wood cylinder, internal transverse fracture from the wood cylinder, and external transverse fracture. Overall median load forces were relatively similar, with the exception of the failure type transverse fracture (internal). Inner lines: medians; boxes: 25th and 75th percentiles; whiskers: maximum and minimum values; letters indicate significant differences between median values of maximum loads for different types of root failure (Kruskal–Wallis followed by Mann–Whitney post hoc test; *p* < 0.05). (**B**) Bivariate scatter plot of maximum load (N) and root diameter (mm) according to types of root failure of *Selenicereus setaceus*. There is a moderate correlation between larger diameter roots and larger loads. T-shaped and L-shaped types of failure require larger forces than transverse, notably in root diameters greater than approximately 1 mm. Spearman’s correlations: all dataset (n = 75) Rs = 0.77, *p* < 0.001; T-shaped fracture (n = 33) Rs = 0.76, *p* < 0.001; L-shaped fracture (n = 16) Rs = 0.73, *p* < 0.01; transverse fracture (internal) (n = 14) Rs = 0.68, *p* < 0.01; transverse fracture external (n = 12) Rs = 0.59, *p* < 0.01.

## Data Availability

Data supporting the findings of this study are available at Zenodo. https://doi.org/10.5281/zenodo.7868179; Rowe et al. cactus data.xlsx (First access 26 April 2023).

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
