# Peer review of "Failure without Tears: Two-Step Attachment in a Climbing Cactus"

_biomimetics, 2023, doi:10.3390/biomimetics8020220_

Round 1

Reviewer 1 Report

This paper describes a two-steps attachment strategy in the climbing cactus Selenicereus setaceus. The two-steps attachment strategy of climbing cactus was introduced from the authors in Soffiatti, P., & Rowe, N. P. (2020). Mechanical innovations of a climbing cactus: functional insights for a new generation of growing robots. Frontiers in Robotics and AI7, 64. In tropical environment, the climbing cactus provides a quite interesting attachment mechanism which could be mimicked to develop new smart materials for attachment in in complex and unstructured 3D environments. 

Here, Rowe et al., have investigated in detail the biomechanics, functional anatomy and failure mitigation mechanism of "hook and root" attachment strategy of climbing cactus.

The team has tested the maximal strength of spines and roots in cactus plants and characterized their types of failures. The team has also characterized the cross-sections of spines showing interesting transition between the base of the spines and the internal soft tissues. Moreover, the team has characterized the sections of roots showing the root emerging from the skin with interesting visible periderms formation at its point of ruptures.

Overall, the paper is well-written, methodology and results are clearly reported and discussed.  I suggest to accept this paper after minor revisions.

Minor comments:

- Please check English language and style mistakes. Some examples: line 44: for technical functionality; line 80: rainforest; etc

- It could be interesting for future investigations to characterize not only the maximal strength but also the force achieved by pulling the spines at different angles, similarly to Bauer, Georg, et al. "Always on the bright side: the climbing mechanism of Galium aparine." Proceedings of the Royal Society B: Biological Sciences 278.1715 (2011): 2233-2239.; Fiorello, Isabella, et al. "Climbing plant‐inspired micropatterned devices for reversible attachment." Advanced Functional Materials 30.38 (2020): 2003380.
- Materials and methods section: please add the dimensions of the loop used to test the strength of spines (i.e. length and thickness of the loop). 

Author Response

Minor comments:

- Please check English language and style mistakes. Some examples: line 44: for technical functionality; line 80: rainforest; etc

***Thank you, all of these errors have now been, corrected.

- It could be interesting for future investigations to characterize not only the maximal strength but also the force achieved by pulling the spines at different angles, similarly to Bauer, Georg, et al. "Always on the bright side: the climbing mechanism of Galium aparine." Proceedings of the Royal Society B: Biological Sciences 278.1715 (2011): 2233-2239.; Fiorello, Isabella, et al. "Climbing plant‐inspired micropatterned devices for reversible attachment." Advanced Functional Materials 30.38 (2020): 2003380.

***Yes, thank you. We have mentioned this fact in the text and cited the two articles concerned. We made a comment mentioning the usefulness of this and at the same time stated that this more detailed approach would was not really achievable in our field work constraints but would clearly be an interesting further study.

- Materials and methods section: please add the dimensions of the loop used to test the strength of spines (i.e. length and thickness of the loop).

***Thank you , we have added more specific details concerning the fishing line material, the loop dimensions and the set up in general and why we chose this for the field measurements.  

Reviewer 2 Report

This paper proposes a two-step attachment of a climbing cactusa. It could bring inspirations for design of climbing robots. However, the main idea of the paper is not clear enough, and it is hard for reader to fully understand. The following comments may be helpful in improving the manuscript:

1. What does ‘two-step attachment strategy’ mean?

2. In Abstract, line 14, ‘One spine can support an average force of 2.6 N which corresponds to an equivalent stem length of 2.6 m (mass of 285g)’, please explain the meaning.

3. What are the differences between spines in English ivy and spines in cactus?

4. In Page 4, line 156, a raised question ‘How are mechanically rigid and strong structures are structurally integrated into a soft body?’ is meaningful. What can we learn from the spine structure of cactus to address that problem?

5. There are some grammatical errors and typos in the paper. Some of them are shown as follows, but not limited to these:

a)  In page 3, line 119, ‘[31, 32’ should be ‘[31, 32]’.

b)  In page 3, line 125, what does ‘{Isnard, 2008 #32]’ mean?

c)  In page 4, line 149, ‘How and are hooks and adhesive roots deployed by the plant in its natural environment?’ should be ‘How are hooks and adhesive roots deployed by the plant in its natural environment?’.

Author Response

  1. What does ‘two-step attachment strategy’ mean?

***We have re-written several lines explaining the essential elements of two step attachment in the abstract.

  1. In Abstract, line 14, ‘One spine can support an average force of 2.6 N which corresponds to an equivalent stem length of 2.6 m (mass of 285g)’, please explain the meaning.

***We have modified the brief description here to explain this. In the abstract.

What are the differences between spines in English ivy and spines in cactus?

***We presume the referee means roots here since there are no spines in Ivy. This is a good point we have added a few lines on this at the end of the introduction.

  1. In Page 4, line 156, a raised question ‘How are mechanically rigid and strong structures are structurally integrated into a soft body?’ is meaningful. What can we learn from the spine structure of cactus to address that problem?

***We have re-phrased this point and added a line or two emphasizing the safe attachment and benign failure that we observe in the biological system.

  1. There are some grammatical errors and typos in the paper. Some of them are shown as follows, but not limited to these:

 In page 3, line 119, ‘[31, 32’ should be ‘[31, 32]’.

***OK thank you this has been fixed.

  1. b)  In page 3, line 125, what does ‘{Isnard, 2008 #32]’ mean?

***OK thank you this has been fixed.

  1. c)  In page 4, line 149, ‘How and are hooks and adhesive roots deployed by the plant in its natural environment?’ should be ‘How are hooks and adhesive roots deployed by the plant in its natural environment?’.

***OK thank you this has been fixed.

***We have checked and corrected all such errors in the manuscript.

Reviewer 3 Report

This is a great study, the only point I would like to raise is the creation and inclusion of a graphical schematic depicting the two-step attachment mechanism described in the text. This will make the concept relevant to a much wider, interdisciplinary audience.

Author Response

This is a great study, the only point I would like to raise is the creation and inclusion of a graphical schematic depicting the two-step attachment mechanism described in the text. This will make the concept relevant to a much wider, interdisciplinary audience.

Thank you this is a good suggestion; we provide a modified diagram based on that in Soffiatti and Rowe 2020 to illustrate the principle again for this paper.

Round 2

Reviewer 2 Report

The revised manuscript has been improved, the following comments may be helpful in improving the manuscript:

(1) All the figures are not  visible in the manuscript.

(2) Some necessary explanations should be added to the video.

Author Response

1) We have now contacted the Journal and it is not necessary to correct any information concerning the missing figures.

2) We have now inserted some text to accompany the supplemental video 1 which reads as the following:

Video S1. Spine attachment in a wind-prone habitat. The Selenicereus setaceous grows in several coastal habitats including wind-prone, shrubby vegetation along the coast of eastern Brazil. This habitat experiences almost constant wind-blown conditions. In these conditions, the cactus depends on the deployment of recurved spines (step 1 attachment) to anchor the climbing stems among dense branches of shrubs.

3) Following the referee's comments from round 1 asking for some of the english to be made more accessible, we have modified the last sentences of the abstract (please see attached file with track changes on Page1). The following text replaces the last lines from line 22:

In this cactus, the first step deploys hooks that attach to a substrate; this process is instantaneous and is highly adapted for moving environments. The second step involves more solid root attachment to the substrate involving slower growth processes. We discuss how initial fast hook attachment can steady the plant on supports allowing for the slower root attachment. This is likely to be important in wind prone and moving environmental conditions. We also explore how two-step anchoring mechanisms are of interest for technical applications particularly for soft-bodied artefacts, which must safely deploy hard and stiff materials originating from a soft compliant body.